# In Situ Polymerization of Linseed Oil-Based Composite Film: Enhancement of Mechanical and Water Barrier Properties by the Incorporation of Cinnamaldehyde and Organoclay

**DOI:** 10.3390/molecules27228089

**Published:** 2022-11-21

**Authors:** Rim Guesmi, Nasreddine Benbettaieb, Mohamed Ramzi Ben Romdhane, Thouraya Barhoumi-Slimi, Ali Assifaoui

**Affiliations:** 1Department of Chemistry, Laboratory of Structural (Bio)Organic Chemistry and Polymers, Faculty of Sciences of Tunis, University of Tunis El Manar, Tunis 2092, Tunisia; 2UMR PAM A 02.102, Agrosup Dijon, University of Bourgogne Franche-Comté, F-21000 Dijon, France; 3Laboratory of Composite Materials and Clay Minerals, National Center of Researches in Material Sciences (CNRSM), BP 73, Soliman 8027, Tunisia; 4High Institute of Environmental Science and Technology, Technopark of Borj Cedria, University of Carthage, Carthage 1054, Tunisia

**Keywords:** packaging, linseed oil, organoclay, cinnamaldehyde, films, radical polymerization, composite

## Abstract

Linseed oil-based composite films were prepared with cinnamaldehyde (Cin) using a modified clay (organoclay) through in situ polymerization, which is the result of the interaction between Cin and organoclay. The incorporation of organoclay reduces the polymer chain’s mobility and, therefore, increases the thermal stability of the composite films. In some experimental conditions, the clay is located both inside and on the surface of the film, thus, affecting the mechanical and thermal properties as well as the surface properties of the composite films. The incorporation of organoclay decreases the water contact angle of the composite film by more than 15%, whatever the amount of cinnamaldehyde. However, the incorporation of cinnamaldehyde has the opposite effect on film surface properties. Indeed, for the water vapor permeability (WVP), the effect of cinnamaldehyde on the film barrier properties is much higher in the presence of organoclay. The incorporation of hydrophobic compounds into the polymer films reduces the water content, which acts as a plasticizer and, therefore, decreases the WVP by more than 17%. Linseed oil has a natural antioxidant activity (~97%) due to the higher content of unsaturated fatty acids, and this activity increased with the amount of organoclay and cinnamaldehyde.

## 1. Introduction

With the increasing concern about environmental preservation and the end-life scenarios of petroleum-based materials, the use of vegetable oil-based polymers for the development of composite packaging materials has become an active research area over the past decade [1]. Indeed, vegetable oils and their derived fatty acids are one of the cheapest and most abundant, annually renewable natural resources extracted from various seeds of different natures, such as olive, sunflower, soybean, and linseed oils [2]. Efforts have been made to formulate new composite films based on vegetable oil and egg protein, in order to replace preservatives and the primary packaging of baked sweet food [3]. In general, low mechanical properties and gas barrier (water and O_2_) properties are the main obstacles to the development of bio-sourced materials [4]. These materials have excellent monomers, which can be used in the preparation of advanced materials with interesting potential technological applications. The presence of multiple C=C bonds makes biological oils ideal, thanks to their being natural building blocks for the preparation of several useful polymeric materials. Their conversion into green plastics usually proceeds through either fatty acids C=C bond functionalization or their copolymerization with a variety of alkene commoners [5].

Accordingly, the percentage of unsaturated fatty acids is the base of the chemistry of these materials due to their tendency to polymerize. Indeed, many efforts have been made using unmodified vegetable oils to prepare bio-renewable polymers by thermal, cationic, or radical polymerization methods, taking advantage of the carbon–carbon double bonds in the fatty acid chains [6]. The cationic copolymerization of soybean, corn, tung, and fish oils without any chemical modifications with divinylbenzene or a combination of styrene and divinylbenzene comonomers was studied [7,8]. Moreover, crosslinked bacterial polyesters were obtained by the free radical copolymerization of soybean oil acids. Thus, the radical copolymerization of conjugated linseed oil with styrene and divinylbenzene was studied [9]. However, other works successfully prepared useful bioplastics from tung [10] and linseed oil [9] by thermal copolymerization at relatively high temperatures (200–300 °C).

In this work, we chose regular linseed oil (LO) as a monomer because it can polymerize easily due to its high content of unsaturated fatty acids (linolenic acid representing 53%) [9]. Linseed oil was polymerized in the presence of cinnamaldehyde (Cin), which is the main component of cinnamon oil. Cinnamon oil has antimicrobial, antifungal, and antioxidant activities [11,12]. Indeed, this aromatic essential oil could be used at low concentrations, thus, minimally altering organoleptic properties [13]. Radical copolymerization between cinnamic monomers such as cinnamaldehyde, methyl cinnamate, cinnamic acid, N-isopropyl cinnamide, cinnamaldehyde, and cinnamonitrile, and other vinyl monomers such as methyl acrylate and styrene, was studied in conventional radical polymerization [14]. However, these materials suffer from low mechanical, thermal and water barrier properties compared to pure polymer [15,16]. To improve these properties, some research works incorporated clay during the formulation process [17,18]. For example, an active packaging film was prepared by the incorporation of thyme oil with natural Na-montmorillonite and organo-montmorillonite in the poly-L-lactic-acid matrix. Indeed, the authors showed significant amelioration in the water vapor permeability (WVP) and mechanical properties [19]. Clays, used as reinforcing elements in composite materials, have been the subject of numerous studies over the last decades. Several types of clay, such as montmorillonite, illite, kaolinite, and vermiculite are used as fillers [20]. Clays are phyllosilicates composed of octahedral aluminum layers and tetrahedral silicate layers with a hydrated characteristic. The interlayer space is composed of exchangeable inorganic cations such as Na^+^ and Ca^2+^, which are linked by weak forces [21]. Among these, montmorillonite (MMT) is the most preferred nano-filler due to its largest surface area and highest cation exchange capacity [22]. Clay is an inexpensive natural mineral that has been used as filler for rubber and plastic materials for many years, but its reinforcing ability is poor, so it can only be used for conventional micro-composites. To improve its reinforcing ability, clay can be chemically modified with some surfactants to be compatible with organic monomers and polymers [23]. Polymer clay composites exhibit three distinct morphologies: conventional segregated clay phase and intercalated and exfoliated structures. The degree of clay exfoliation and intercalation depends on the type of clay, cation exchange capacity, chemical nature of the interlayer cations, curing agent, curing temperature, and time and dispersion method [24]. Several processing routes can lead to well-dispersed layered silicate nanocomposite: in situ intercalative polymerization, solution polymerization, and melt processing [25].

In the present work, linseed oil-based composite films containing both cinnamaldehyde (Cin) and organoclay were synthesized with in situ intercalative polymerization. The principle of this synthesis is to perform the polymerization directly between the clay sheets. First, the natural clay (montmorillonite) was modified by CTAB (Cetyl Trimethyl Ammonium Bromide) to form the organoclay (OMMT). Then, the film composite was made by reacting the OMMT in different amounts with cinnamaldehyde-covered linseed oil-based prepolymer. The synthesized composite films were then characterized by various techniques such as thermal gravimetry analysis (TGA), differential scanning calorimetry (DSC), X-ray diffraction (XRD), Fourier transform infrared (FTIR) spectroscopy, and scanning electronic microscopy (SEM). The water and gas barriers as well as the surface and mechanical properties of the composite film were also tested. Finally, the antioxidant activity was tested using the DPPH assay. Note that the composite dispersion is designated as C_x_OM_y_ (where C and OM indicate the cinnamaldehyde and the organoclay, respectively).

## 2. Results and Discussion

### 2.1. Clay Modification

The FTIR spectra of both MMT and OMMT are presented in Figure 1a. For the two samples, peaks at 3428 cm^−1^ and ~1636 cm^−1^ were observed and assigned to OH stretching with the hydrogen bending in the interlayer space and the H-O-H bending. The peak at 1036 cm^−1^ can be attributed to the Si–O–Si stretching vibrations, which demonstrates the structure of a siloxane linkage, and the band around 3691 cm^−1^ is attributed to the OH stretching vibration of Al-OH. For the organoclay, new absorption peaks at 2932 and 2850 cm^−1^ were observed and assigned to symmetric and asymmetric vibrations CH stretching present in the alkyl chain of the surfactant used to modify the clay cetyltrimethylammonium bromide (CTAB)). The band at 1471 cm^−1^ was assigned to the N-R_4_ ([CH_3_ (CH_2_)_15_ N^+^(CH_3_)_3_], Br^−^) vibration of CTAB [26]. Finally, the FTIR results clearly indicate that the CTAB cations exchanged with the sodium ions of montmorillonite clay were intercalated in the galleries [27].

X-ray diffractograms of MMT, Na^+^MMT, and OMMT are shown in Figure 1b. A strong diffraction peak at 2*θ* = 6.6° for MMT corresponds to a Bragg spacing (d_001_) of 14.08 Å, the interlayer spacing of the (001) MMT plane. XRD also shows the presence of a high-intensity peak at d_001_ = 7.12 Å of Kaolinite, indicating that this species was present in the sample [28]. The Na^+^-MMT model showed a broad peak at d_001_ = 12.17 Å, indicating that the cation exchange was successful, while the OMMT pattern presented a strong diffraction peak at 2θ = 3.4°, which corresponds to a Bragg spacing (d) of 25 Å in the interlayer spacing of the (001) plane of OMMT. The increase in the interlayer spacing of the clay was due to the intercalation of organ chains (alkyl ammonium) between the clay layers [29]. Therefore, it can be said that the increase in interlayer spacing shows that the sodium in montmorillonite was replaced by CTAB cations in the aluminosilicate layers [30].

### 2.2. Film Composites: Morphology, Mineral Composition, Physical States, and Structural Features

#### 2.2.1. Thermal Analysis (TGA and DSC)

The synthesized composite films C_5_OM_0_, C_5_OM_3_, and C_5_OM_5_ contained 5 wt% of cinnamaldehyde and various amounts of clay (0, 3, and 5 wt%). To estimate the real amount of the inorganic (aluminosilicates) in the composite films, the thermal gravimetry analysis (TGA) is first studied (Figure 2). When organoclay was incorporated from 0 to 5wt% into the film matrix, a shift in the degradation temperature (from 411 °C to 428 °C) was observed. For C_5_OM_0_, the degradation above 400 °C was mainly due to the linseed oil chain scission (the crosslinked networks). About 6% of residues remained, due to the scission of the linseed oil chain, which did not completely degrade, since the TGA analysis was performed under nitrogen N_2_ and not under O_2_. In the C_5_OM_5_ sample with 5% clay, about 17% residue remained at the end of the experiment. The remaining mass was due to the decomposition of organic molecular chains from organoclay and remnant polymer linseed oil chains [31]. These analyses show that the increase in the content of organoclay improves the thermal stability of composites.

DSC was used to study the ageing phenomenon of the films by long-time storage (17 h) at temperatures slightly below the Tg value (glass transition temperature), corresponding to the glassy states of the composite films. During the first heating (after the ageing step) an endothermic peak was observed for the four composite films (Figure 3). This peak corresponds to the melting of the crystalline forms which are formed during physical ageing. The storage of the composite films at T ≤ Tg allows the system to organize, and thus, crystallize. The second heating allows erasure of the thermal history of the film and is generally used to determine the glass transition temperature. The second and third heatings of the various composite films (data not shown) did not show any significant changes in the specific heat, and we were unable to determine any Tg, probably due to its large temperature range. However, this ageing temperature seems to evolve in the same way as Tg values [32]. For C_5_OM_0_, it must be noted that the temperature of storage was equal to −35 °C and the endothermic peak was observed at −17 °C. At this temperature, the polymer chains were mobile enough to organize themselves and crystallize (molecular relaxation). This melting temperature shifted to a high temperature when the amount of clay was increased. Thus, the temperature of the peak was equal to −17, −16, −11, and −9 °C for C_5_OM_0_, C_5_OM_1_, C_5_OM_3_, and C_5_OM_5_, respectively. This shift can be explained by the large intercalation of organoclay in the film matrix, which reduced the polymer chain movement on the composite structure. Shekarabi et al. [33] displayed a significant increase in Tg values with increasing organoclay in quince seed mucilage edible films. Organoclay particles create strong bonds between polymer chains, which limit the polymer chain’s mobility, reduce the free volume, and therefore, require higher temperatures to induce the movement of chains. The thermal stability of the film is also improved after organoclay incorporation. The increase in Tg and thermal degradation temperature (Figure 2) explains the densification of the polymer network, and therefore, the improvement of water barrier properties and tensile strength previously discussed when organoclay is added.

#### 2.2.2. Structural Studies (FTIR)

The FTIR spectra for the linseed oil and its composites are presented in Figure 4. In the spectrum of linseed oil (LO), strong signals were found at 1745 and 1160 cm^−1^, which were attributed to C=O and C-O stretching of the fatty acids, respectively. The bands at 3015 and 718 cm^−1^ were from the stretching vibrations of the double bonds, =C-H, C=C, and cis-CH=CH, respectively (Figure 4a). In addition, the peaks at 2930 and 2860 cm^−1^ were attributed to the asymmetric and symmetric CH stretching vibrations of the methylene groups, while their asymmetric and symmetric bending vibrations occurred at 1462 and 1378 cm^−1^, respectively. For C_5_OM_0_, the bands at 3015 and 718 cm^−1^ were not visible, indicating the disappearance of double bonds during the free radical polymerization reaction. The spectrum of the composites was quite similar to that of the polymerized C_5_OM_0_, with new peaks attributed to Si-O-Si at 1036 cm^−1^. After the incorporation of the clay, the intensity of the C=O and Si-O-Si stretching bands increased considerably from 1 to 5%(*w/w*), while during curing, the bands related to the carbon–carbon double bonds disappeared, indicating the onset of the cross-linking process (Figure 4b).

#### 2.2.3. Physical State of the Films and Microscopy

XRD measurements were performed on C_5_OM_0_ and C_5_OM_5_ (Figure 5). The two film composites exhibited the same pattern with a broad peak at 20°, which appeared due to the amorphous nature of the film. The disappearance of the peak due to organoclay in C_5_OM_5_ confirmed the distortion of the platy monolayers of the clays [34].

Scanning electronic microscopy showed a homogenous surface in C_5_OM_0_ (Figure 6a), while in C_5_OM_5_, mineral particles were observed on the surface of the film, as confirmed by their elementary analysis (see Appendix A). We can suppose that most of the aluminosilicate (organoclay), which is in the amorphous form (Figure 5c), was imbibed in the organic film.

### 2.3. Film Composites: Functional Properties

#### 2.3.1. Water Barrier Properties

Figure 7 displays the water vapor permeability (WVP) (33–84%) RH gradient of composite films containing different amounts of cinnamaldehyde and organoclay. First, in the absence of cinnamaldehyde, high WVP was obtained (~8 × 10^−11^ g/m·s·Pa). The presence of 3% of the organoclay (C_0_OM_3_) slowly decreased the WVP compared to the control film (C_0_OM_0_). In the absence of the organoclay, the WVP decreased significantly (about four times) when 5% of cinnamaldehyde was introduced in the composite film (C_5_OM_0_), compared to the C_0_OM_0_: 8.33 ± 2.94 × 10^−11^ for C_0_OM_0_ film to 2.96 ± 1.34 × 10^−11^ g/m·s·Pa for C_5_OM_0_ film. This decrease could be explained by the hydrophobic character of cinnamaldehyde, which reduces water transport.

The effect of cinnamaldehyde on the film barrier properties was much higher in the presence of organoclay, and the WVP decreased by 75% after cinnamaldehyde incorporation at 5% (C_5_OM_3_) to the film with 3% organoclay (C_0_OM_3_). In general, the incorporation of hydrophobic compounds into the hydrophilic polymer films reduces the moisture content, which is responsible for the plasticity of the film, since water acts as a plasticizer, and therefore, increases the WVP [35]. The decrease in the WVP observed for C_5_OM_3_ could be explained by the hydrophobic character of free Cin. Moreover, some Cin molecules participated in the polymerization reaction (Figure 8), likely increasing the polymer network density, which may have reduced the water transfer.

Some authors observed a decrease in WVP in proteins and polysaccharides-based films after the incorporation of cinnamaldehyde. In sodium alginate/carboxymethylcellulose films, Han et al. (2018) [36] showed that cinnamaldehyde (a non-polar substance) effectively inhibits vapor diffusion and, subsequently, improves the water vapor barrier. Similarly, Srisa et al. (2020) [37] reported that the addition of trans-cinnamaldehyde up to 5% significantly decreased the WVP of poly(butylene adipate-co-terephthalate)/Poly(lactic acid) (PBAT/PLA), while 10% trans-cinnamaldehyde greatly increased WVP due to the plasticization effect, later leading to the increased WVP of the blend films. We can conclude that films containing both organoclay and cinnamaldehyde (C_5_OM_1_, C_5_OM_3_, C_5_OM_5_) have lower WVP than that of the control film (C_0_OM_0_), but without any significant effect of organoclay on the WVP of film at 5% cinnamaldehyde. In contrast, Pirsa et al. (2020) [38] showed a higher decrease in the WVP of pectin film incorporated with nanoclay. Naidu et al. (2020) [39] displayed a 48% decrease in WVP of the xylan-alginate film after the addition of natural clay (MMT) at 5% wt. They explained this decrease by the creation of a more complex, challenging, and longer path for water molecules to travel through, thereby reducing the amount of water travelling through the film as compared to the unfilled xylan-alginate matrix. In addition, the clay in the film matrix creates a discontinuous zone, and thus, prevents the mobility of polymer chains. The reduced mobility of polymer chains induced by the nanoparticles might enhance the barrier properties of films. The same trend was reported by Giannakas et al. (2022) [19] after the addition of organoclay in the PLA matrix. Therefore, the WVP decreases by more than 14% after the addition of 5% of organoclay. In our case, the effect of clay was masked by the hydrophobic character of linseed oil and cinnamaldehyde content, which play a major role in the water transport through the film network.

#### 2.3.2. Mechanical Properties

The mechanical properties of linseed oil films with and without cinnamaldehyde and organoclay at different concentrations are given in Table 1. The tensile strength (TS) of film at both 5% cinnamaldehyde and 0% organoclay (C_5_OM_0_) was about 0.8 MPa. When organoclay was incorporated, the TS value significantly (*p* ˂ 0.05) increased for C_5_OM_1_ and C_5_OM_3_ (6.5 and 9.8 MPa, respectively) and reached a plateau for C_5_OM_5_ (9.8 MPa). Young’s modulus (YM) significantly (*p* ˂ 0.05) continuously increased from 4 MPa for C_5_OM_0_ to 156, 256, and 735 MPa for C_5_OM_1_, C_5_OM_3,_ and C_5_OM_5_, respectively. On the other hand, the value of elongation at break (E, %) significantly decreased from 26.2% for C_5_OM_0_ to 12.8, 6.4, and 1.1% for C_5_OM_1_, C_5_OM_3,_ and C_5_OM_5_ films, respectively. The same tendency was observed in the absence of cinnamaldehyde (C_0_OM_0_ and C_0_OM_3_). Indeed, the presence of organoclay decreased the elongation at break and increased both the TS and YM. This increase in TS and YM after organoclay addition on linseed oil films could be explained by the good dispersion of clay on the film matrix and the creation of a more reinforced phase within the matrix. It seems that the addition of clay improves film strength due to the strong interaction between clay and linseed oil, resulting from a reduced-free volume and molecular mobility of the polymer. This is in agreement with the results reported by Naidu et al. (2020) [39] in the case of xylan-alginate films, who found that by incorporation of 5 wt% of clay (bentonite), tensile strength and Young’s modulus are improved by 112% and 76%, respectively, compared to control film without clay. The same behavior was reported by De Silva et al. (2013) [40], who found that the addition of halloysite nanoclay at 5 (*w/w*%) on chitosan films improved TS and YM by 34% and 21%, respectively. A similar observation was reported by Ortiz-Zarama et al. (2016) [41] and Lee et al. (2018) [42], who displayed that the addition of bentonite or halloysite nanoclay, respectively, to gelatin and chitosan films, increased the TS and decreased the elongation at break of the film.

In previous studies, the TS of chitosan composite films increased with the amount of nanoclay particles, owing to the possible strain-induced alignment of the clay particle layers in the polymer matrix and intermolecular interactions (hydrogen bond) between the polymer and clay mineral [43]. The stretching resistance of the oriented backbone of the polymer chain during the polymerization process also contributed to enhancing the modulus and the stress. The deformation decreased with the organoclay content because the layered organoclay acts as a mechanical reinforcer of the film matrix, therefore, reducing their flexibility. In the absence of the organoclay, when cinnamaldehyde is incorporated in the film (C_0_OM_0_ and C_5_OM_0_), a slight decrease in TS and YM accompanied by an increase in E% are observed. In the presence of 3% of organoclay (C_3_OM_3_ and C_5_OM_3_), a huge increase in the YM and the TS was observed compared to the films without clay. We can conclude that the mechanical properties are governed by the presence of the organoclay. In addition, the increase in cinnamaldehyde in (C_3_OM_3_ and C_5_OM_3_) from 3% to 5% did not affect the mechanical properties.

Han et al. (2018) [36] observed a decrease in TS by 30% and an increase in E (%) by 60% after the addition of cinnamon essential oil to sodium alginate/carboxymethylcellulose films. They reported that oil reduces the cohesion forces on the polymer network, and therefore, increases the mobility of the chains, which increases the extensibility and flexibility of the films. This effect is related to an increase in the porosity of the film after drying due to the oil droplet’s presence. The same trend was reported by Wu et al. (2018) [44] in the case of soy protein isolate film by incorporating plant-sourced cinnamaldehyde.

#### 2.3.3. Surface Properties

The water contact angle (WCA) of composite films was calculated with various amounts of organoclay (OMMT) and cinnamaldehyde. The surface free energy (YS), and their respective polar (YSP) and dispersive (YSD) components are presented in Table 1. The WCA is known as an indicator for the degree of surface hydrophobicity/hydrophilicity and wettability of the films. Generally, films with a WCA higher than 65° are considered hydrophobic surfaces [45]. All formulated films showed a higher degree of WCA, proving their hydrophobic nature principally related to the presence of a high amount of linseed oil. WCA film with 5% cinnamaldehyde (C_5_OM_0_) decreased after the incorporation of organoclay at 1, 3, or 5%; inversely, the polar component and surface free energy increased. The modified organic chain of clay stratifies the surface during the film formation and crosslinking, which increases the surface free energy. The introduction of OMMT changes the surface roughness, which influences the contact angle and surface free energy values. This agrees with microscopy observations showing that a part of the clay was located on the film surface (Figure 6). The increase in the surface free energy of the films was assisted by the polar component rather than by the dispersive part. The improvement of the polarity after organoclay addition was attributed to the interaction of this later with C=C groups of linseed oil during the polymerization process, and therefore, the reduction in film matrix interaction. This behavior was confirmed by Yoksan et al. (2010) [46] in the case of chitosan-starch film incorporated with nanoparticle-loaded silver. Cyrasa et al. (2008) [47] reported a significant decrease in the dispersive component of surface free energy but an increase in the polar component with the addition of OMMT to the thermoplastic starch nanocomposite films. This behavior could be due to the hydroxylated silicate layers that increase the composite’s hydrophilicity. In the absence of organoclay, the water contact angle of the film (C_0_OM_0_, 86.9 ± 4.3°) remained in the same range as that observed after cinnamaldehyde incorporation at 5% (C_5_OM_0_) (88.3 ± 3.6°). However, at 3% organoclay, the WCA increased with cinnamaldehyde content; inversely, polar component decreased. This phenomenon was evident and mainly explained by the hydrophobic character of cinnamaldehyde, which increased the surface hydrophobicity.

### 2.4. Antioxidant Activity (AA%)

The antioxidant activity (AA, %) was assessed using the method based on the scavenging of the DPPH• radical molecule. The AAs of the composite films with various amounts of organoclay (OMMT) and cinnamaldehyde are displayed in Table 2. Firstly, the natural antioxidant activity of the pure compounds was evaluated. Linseed oil has a higher AA (81.5 ± 2.5 and 97.1 ± 0.1, respectively, after 2 and 24 h). This high value of AA is related to its higher fatty acid content. The GC-MS analysis of linseed oil (data not shown here) displayed that this oil has three principal fatty acids: 22% of C18:1, 15% of C18:2, and 50% of C18:3. The higher amount of C18:3 in linseed oil is responsible for its higher AA. This is in accordance with the work of Symoniuk et al. (2017) [48], which displayed that polyunsaturated fatty acids had the largest share in the fatty acid composition of linseed oil, and the most abundant was α-linolenic acid (around 53% content), which is most responsible for the oxidative stability of linseed oil. Barthet et al. (2014) [49] reported the same content of α-linolenic acid (57%) in linseed oil. In the same work, the authors added that the higher α-tocopherol content in linseed oil explains its higher antioxidant activity evaluated by different methods (DPPH, ORAC, etc.). In addition, linseed oil content, polar lipids such as glycolipids and phospholipids, as well as some phenolic acids (ferulic acid) [48], might also have contributed to its higher antioxidant property. As reported by several studies, linseed oil showed higher antioxidant activity in the form of hydroxyl and superoxide radical scavenging [50,51,52].

Cinnamaldehyde alone had a very low AA activity (less than 2%) after 2 h and slowly increased after 24 h, without any significant difference between the three concentrations used (1, 3, and 15 mg/10 mL DPPH solution). In general, cinnamaldehyde has good scavenging activity, but in the concentration used here, its activity was very low.

The AA of organoclay was higher than cinnamaldehyde at the same concentration used (6%) after 2 h. This AA significantly increased with the concentration of the organoclay and with the contact time with the DPPH solution. It must be highlighted that the AA of cinnamaldehyde and organoclay was still lower compared to pure linseed oil, which was due to their very low content in the films. The free radical scavenging activity of organoclay is basically due to the presence of alkylammonium groups in the structure after modification of clay by the CTAB, and the antioxidant activity may be due to the transfer of an electron from the N-R group to the odd electron located at the nitrogen atom in DPPH, which induces an increase in AA. As seen in Table 3, the control film (C_0_OM_0_) had higher AA (52.6 ± 2.8% at 2 h), which significantly increased after 24 h (94.6 ± 1.3%). This is related to the natural AA of linseed oil, as previously discussed. The AA of film with 5% cinnamaldehyde and 0% organoclay (C_5_OM_0_ film) remained in the same range as the AA of the control film (C_0_OM_0_). The AA decreased from 58 ± 2.6% to 28.7 ± 0.3% after 2 h and from 90.5 ± 0.8% to 70.7 ± 0.5% after 24 h, when organoclay (at 1%) was incorporated into the linseed oil film at 5% cinnamaldehyde. This decrease in AA after the incorporation of organoclay at 1% was mainly related to the later neutralization effect of this on the free radical of linseed oil, therefore, inhibiting the contact between the DPPH free radical and oil-free radical. Thus, 1% of clay was sufficient to crosslink the matrix. Then, when the concentration of organoclay increased to 3 and 5%, the AA significantly increased, reaching 49.0 ± 1.1% and 82.1 ± 0.5%, respectively, at 2 h and 24 h for C_5_OM_5_ film. This increase was related to the scavenging activity of organoclay, as previously discussed. This was confirmed by Dairi et al. (2019) [53], who displayed that organoclay increases the AA activity of cellulose acetate and thymol nano-biocomposite films. The incorporation of cinnamaldehyde at 1, 3, and 5% in the linseed oil film with 3% organoclay did not affect the AA of the films. As previously reported from the activity of pure compounds, the cinnamaldehyde at these concentrations had very low AA, and therefore, the activity of these films was related to the natural activity of linseed oil. On the contrary, López-Mata et al. (2018) [54] displayed that the incorporation of 1% cinnamaldehyde into chitosan films was enough to increase the antioxidant activity. They showed that the AA of cinnamaldehyde was related to the hydrogen group in their structure. In that case, they can play the role of an electron or hydrogen donor, which could react with free radicals to convert them into more stable products and terminate the radical chain reaction, but to a lesser extent than linseed oil with a higher fatty acid content.

## 3. Materials and Methods

### 3.1. Materials and Reagents

The flaxseeds were purchased from the market. Flaxseed oil (Linseed oil) was extracted by the Soxhlet method using hexane as a solvent. Cinnamaldehyde (Cin), sodium chloride, ethanol, and hexane were purchased from Aldrich Chemical Company (St. Louis, MS, USA). Cetyl trimethyl ammonium bromide (CTAB) (99% purity) is a cationic surfactant with an average molecular weight of 364 g·mol^−1^, and this was purchased from Aldrich Chemical Company. Peroxide benzoyl 98% was used as a curing agent and was purchased from Aldrich Chemical Company. Clay montmorillonite MMT was collected from the region of Oued El Abid (northeast of Tunisia). Potassium chloride (KCl) from Sigma-Aldrich (Chimie SARL, Saint-Quentin Fallavier, France) was used to prepare the saturated salt solutions to fix the relative humidity (at 84%) for the water vapor measurement.

### 3.2. Methods

#### 3.2.1. Extraction of Linseed Oil

The flaxseeds were cleaned and the contaminants eliminated; then, they were dried in the open air for 3 days, followed by drying in the oven at 103 °C for 48 h. Finally, they were crushed. An amount of 40 g of flax seeds of *L. usitatissimum* was ground to a fine powder and then macerated with 300 mL of hexane at 60 °C for 6 h. Several extractions were carried out for different durations in 300 mL of hexane to obtain the optimal duration that would produce the highest yield (42%). The obtained mixture was filtered through filter paper (1 µm of the pore), concentrated further in a vacuum at 40–50 °C, and stored at 20 °C until analysis.

#### 3.2.2. Modification of the Montmorillonite (OMMT)

Sodium montmorillonite (Na^+^MMT) clay was obtained by treatment with diluted HCl (100 mM) to remove carbonates, saturated five times with NaCl (1 M) solution, and washed with distilled water until it was clear of chloride ion (negative test with silver nitrates, AgNO_3_). After centrifugation at 4000 rpm for 30 min, the clay fraction was collected and dried at 80 °C. The organoclay (OMMT) was prepared by stirring 10% Na^+^MMT suspension with a cationic surfactant (CTAB) at 1 M. The whole dispersion was shaken at room temperature for 48 h. The exchanged clays were centrifuged at 3000 rpm for 5 min many times (5 to 10 times) with distilled water to remove the excess bromide ions. This was checked by using the silver nitrate test, which was negative. The organoclay was dried at 80 °C for 12 h.

#### 3.2.3. Film Preparation

First, the polymeric composites were prepared by mixing various components of linseed oil (LO), cinnamaldehyde (Cin), and organoclay (OMMT) in a glass vial. The composition of the different mixtures is reported in Table 3. The mechanism of the polymerization was as follows (Figure 8): first, the OMMT interacts with Cin through the hydroxyl and ammonium groups of the organoclay and the aldehyde function of Cin to form a stable chain cycle. Such interaction increases the reactivity of the ethylenic carbons of Cin. The addition of peroxide benzoyl (initiator) produces a stable free radical on the Cin molecule (Cin•). The formed OMMT-Cin• then interacts with LO, which is in excess. The polymerization chain reaction continues to be produced until the formation of OMMT-Cin-(LO)x. where x is the number of LO chains forming the polymer.

The mixture is stirred overnight for proper intercalation and then heated at 130 °C under stirring using peroxide benzoyl as a curing agent, allowing an increase in the viscosity of the mixture. In this condition, the fillers are not separated from the polymer when stirring is stopped. The color of the oil changes from light yellow to dark brown as the polymerization time increases. The viscous mixtures are then processed by casting in an appropriate mold (~7 mL by 100 cm^2^) and placed into an oven at 130 °C for 48 h to obtain the composite films. The composite dispersion is designated as C_x_OM_y_ (where C and OM indicate the cinnamaldehyde and the organoclay, respectively).

### 3.3. Film Characterization

#### 3.3.1. Thickness Measurement of Films

The thickness of films was measured in at least 6 different positions using a micrometer (Precision 1 μm, Coolant Proof micrometer IP 65, Mitutoyo, Aurora, IL, USA).

#### 3.3.2. Water Vapor Permeability (WVP)

The water vapor permeability (WVP) was determined according to the gravimetric cup method described in the Standard Test Methods for Water Vapor Transmission of Materials [55]. The relative humidity gradient used in this test was 33–84%. Before WVP measurements, all film samples were equilibrated at 25 °C and 33% relative humidity for 72 h using a saturated saline solution. The film samples (9.08 cm^2^ discs) were placed between two Teflon rings on the top of the glass cell containing a salt solution of KCl (84% RH). The permeation cells were then introduced into a climatic chamber (KBF 240 Binder, ODIL, France) maintained at 33 ± 1% RH and 25 ± 0.5 °C and periodically weighed. The WVP (g·m^−1^·s^−1^·Pa^−1^) determination was based on the change in the absolute value in weight loss of the permeation cell versus time once the steady state was reached and calculated according to the following equation:(1)$$\mathrm{WVP}=\frac{\Delta m}{\Delta t\times A\times \Delta p}\times e$$
where ∆m/∆t is the weight loss per unit of time (g·s^−1^), A is the film area exposed to the moisture transfer (9.08 × 10^−4^ m^2^), e is the film thickness (m), and ∆p is the water vapor partial pressure differential between the two sides of the film (Pa). Δp is equal to the variation of water activity between the two sides of the film (aw_wet_ − aw_dry_) multiplied by the value of the saturated water vapor pressure (3145.6 Pa) at the measuring temperature (25 ± 1 °C). Five replicates were performed for each sample.

#### 3.3.3. Characterization of Surface Properties of Films

The contact angle (θ, °), the surface tension (surface free energy) of films (ᵞ_s_) and their polar (ᵞ_s_^P^) and dispersive (ᵞ_s_^D^) components were determined using the following equation [56]:(2)$${\;\gamma }_{L}\left(1+\cos\theta \right)=2\left(\sqrt{{\;\gamma }_{S}^{D}{\;\times \;\gamma }_{L}^{D}\;}+\sqrt{{\;\gamma }_{S}^{P}{\;\times \;\gamma }_{L}^{P}}\;\right)\;$$

θ, ᵞ_L_, ᵞ_L_^D^, ᵞ_L_^P^ are, respectively, the contact angle, surface tension, and dispersive and polar components of the surface tension of the liquid tested; ᵞ_S_^P^ and ᵞ_S_^D^ are the polar and dispersive components of the surface tension of the film surface tested. The contact angle is expressed in degrees (°), and all surface tension parameters are expressed in mN·m^−1^. Three liquids (water, ethylene, and glycol) with well-known polar ᵞ_L_^P^ and dispersive ᵞ_L_^D^ contributions were used.

By dividing $2\sqrt{{\;\gamma }_{L}^{D}\;}\;$in (2), we obtain (3)
(3)$$\frac{{\;\gamma }_{L}\left(1+\cos\theta \right)}{2\sqrt{{\;\gamma }_{L}^{D}}}=\sqrt{\gamma_{S}^{p\;}}\;.\sqrt{\frac{{\;\gamma }_{L}^{p}}{{\;\gamma }_{L}^{D}}}+\sqrt{\gamma_{S}^{D\;}}\;$$

Then, the above equation can be represented in the linear form:y=ax+b
where
$$y=\frac{\gamma_{L}\left(1+\cos\theta \right)}{2\sqrt{{\;\gamma }_{L}^{D}}}\;x=\sqrt{\frac{{\;\gamma }_{L}^{p}}{{\;\gamma }_{L}^{D}}}\;a=\sqrt{\gamma_{S}^{p\;}}\;b=\sqrt{\gamma_{S}^{D\;}}$$

From this linear presentation, the slope of the graph gives the polar component, and the vertical intercept gives the dispersive component of the film’s surface free energy. The contact angle measurements were carried out using the sessile drop method on a G1 Kruss goniometer (Kruss GmbH, Hamburg, Germany) equipped with image analysis software (Drop Shape Analyzer 30, Kruss GmbH, Hamburg, Germany), according to Karbowiak et al. (2006) [57]. A droplet of each liquid (∼2 µL) was deposited on the film surface with a precision syringe. Then, the method was based on image processing and curve fitting for contact angle measurement from a theoretical meridian drop profile, measuring the contact angle between the baseline of the drop and the tangent at the drop boundary. Five measurements per film were carried out at 25 (±2) °C and a relative humidity of 50 (±1) %. All films were previously equilibrated at 50% RH and 25 °C for 7 days before analysis.

#### 3.3.4. Mechanical Properties

The mechanical properties of the composite films were determined using a universal traction-testing machine (TA.HD plus model, Stable MicroSystems, Haslemere, UK) with a 300 N load cell. Rectangular film samples (2.5 × 8 cm^2^) were cut and equilibrated for 10 days at 25 °C and 50% relative humidity (RH). The equilibrated film samples were then placed in the extension grips of the testing machine and stretched uniaxially at a rate of 50 mm/min until breaking. The initial distance between the two gripes was fixed at 50 mm. From the stress–strain curves, the tensile strength (TS, MPa), Young’s modulus (YM, MPa), and percentage of elongation at the breakpoint (E, %) were determined, according to NF 527-3 EN (ISO 1995) [58]. Measurements were carried out at room temperature (25 ± 2 °C), and five samples for each formulation were tested.

#### 3.3.5. X-ray Diffraction (XRD)

The polymer composite films, pure clay (MMT), and organoclay (OMMT) were deposited in stainless steel sample holders. The X-ray diffraction (XRD) patterns were collected with a Siemens D5000 diffractometer using Cu Ka (λ = 1.5406 Å) radiation and an INEL CPS 120 curved detector.

#### 3.3.6. Scanning Electron Microscopy (SEM)

The surface morphologies of the films were observed by scanning electron microscopy using a Hitachi SU-1510 microscope with an accelerating voltage of 20 keV. The films were coated with a carbon layer to increase the quality of the images.

#### 3.3.7. Fourier Transform Infrared (FTIR) Spectroscopy

FTIR was used to study the various interaction involved between linseed oil and OMMT. This analysis was performed using a Perkin Elmer 65-FTIR spectrometer (Haguenau, France) equipped with an ATR accessory with a ZnSe crystal. The IR spectra were collected in the 4000 to 400 cm^−1^ range, with 16 scans and with a resolution of 2 cm^−1^. Triplicate measurements from the same sample were carried out at room temperature (25 °C).

#### 3.3.8. Differential Scanning Calorimetry (DSC)

The thermal properties of the composite films were conducted by differential scanning calorimetry (DSC, METTLER TOLEDO DSC 03 model STAR^e^System, Mettler Toledo, Columbus, OH, USA). It must be noted that at room temperature, all the composite films studied were flexible, indicating that the glass transition temperature (Tg) was negative. It must be also highlighted that the determination of the Tg in polymeric films is difficult, since it may occur at a larger temperature range and has low specific heat [32]. We first stored a piece of the composite film in a freezer at different ageing temperatures (T_ageing_) (−35, −30 and −25 °C), and we observed its physical state (solid or soft) to visually determine the range of its Tg. Thus, for C_5_OM_0_, we observed that at T_ageing_ = −35 °C, the film became hard and brittle, indicating that its Tg ranged between this temperature and the room temperature. The composite film (6–7 mg) was introduced into the calorimeter under a nitrogen atmosphere, and a thermal treatment was applied (Figure 9). First, the sample was heated to 40°C to erase its thermal process history (1); then, it was kept at T_ageing_ value for 17 h (2) to allow the film to crystallize (physical ageing). Following this, two successive heats ((3) and (4)) were performed (Figure 9). The aging temperatures equaled −35 °C for C_5_OM_0_ and C_5_OM_1_, −30 °C for C_5_OM_3_, and −25 °C for C_5_OM_5_. The Tg (°C) for each sample was determined from the second heating cycle using TA universal analysis 2000 software (version 4.5 A, TA Instruments, New Castle, DE, USA).

#### 3.3.9. Thermogravimetric Analysis (TGA)

Thermogravimetric studies were carried out by using thermogravimetric under nitrogen (20 mL.min^−1^). The samples were heated from 30 to 600 °C at a heating rate of 20 °C·min^−1^.

### 3.4. Antioxidant Efficacy Testing (DPPH Test)

The DPPH^•^ (2, 2-diphenyl-1-picryl-hydrazyl-hydrate) method is widely used to evaluate the free radical scavenging ability of natural antioxidants or active polymers. The activity of the linseed oil and cinnamaldehyde was assessed based on their ability to scavenge DPPH^•^ radical. This method was described by Benbettaieb et al. (2018) [59] for edible and active packaging films. A piece of film (6 dcm^2^) was introduced into a glass vial containing 10 mL of DPPH^•^ solution (50 mg·L^−1^) in absolute ethanol. This ratio (film surface by volume of food) is determined to be equal to the ratio used for food contact material, according to European standards (6 cm^2^ of packaging by 1 L of food). The glass vial was closed and kept in darkness at 25 °C and under stirring (300 rpm) until the end of the experiment. The antioxidant activity (AA, %) was followed by the disappearance of the DPPH^•^ reactant measured at 515 nm using a UV spectrophotometer (Biochrom WPA Lightwave II UV/Visible spectrophotometer). The purple color of the DPPH^•^ solution became yellow over time, according to the HAT (hydrogen atom transfer), SET (single electron transfer), or mixed mechanisms summarized in the following reaction schemes [60]:
 DPPH·purple at 515 nm + ArOH → DPPHH yellow +ArO·HAT
DPPH·purple at 515 nm + ArOH → DPPH yellow +ArO· ^+^ SET

The AA (%) was calculated according to the following Equation (4)
(4)$$\mathrm{AA}=\frac{A_{\mathrm{blank}}-A_{\mathrm{sample}}}{A_{\mathrm{blank}}}\times 100$$
where A_blank_ represents the absorbance of DPPH^•^ solution without the addition of the film (only DPPH^•^ solution at 50 mg·L^−1^) and A_sample_ represents the absorbance of DPPH^•^ solution containing the film sample. This test was performed in triplicate, and two measurements were carried out after 2 h and after 24 h of the experiment (contact between the film and DPPH^•^ solution). To study the activity of the pure compounds, the AA (%) of linseed oil (300 mg), cinnamaldehyde (3, 9, and 15 mg), and organoclay (3, 9, and 15 mg) in 10 mL of DPPH solution were determined after 2 h and 24 h using the same procedure and calculations. These amounts were chosen to be similar to the amount of these compounds in 6 cm^2^ of films after the drying process, and therefore, to the amount of films tested by the same test, as previously described.

### 3.5. Statistical Analyses

Data were analyzed with a Student’s *t*-test and one-way analysis of variance (ANOVA) (for the groups ˃2) using SPSS 13.0 software (Stat-Packets Statistical analysis Software, SPSS Inc., Chicago, IL, USA). The LSD (least significant difference) mean comparison test was used at the significance level of 95% (*p*-value < 0.05).

## 4. Conclusions

Composite films with linseed oil, cinnamaldehyde, and organoclay are successfully synthesized by in situ polymerization. The organophilic clay is first swollen in a monomer solution, then polymerization is initiated inside the galleries formed by the clay sheets by heating. As the polymerization of monomers occurs, the growing polymer chains pull the sheets apart, thereby individually participating in their dispersion. The synthesis of composite films is facilitated by the presence of cinnamaldehyde. This phenomenon is exasperated by the presence of the organophilic clay OMMT.

Moreover, the addition of clay decreases the synthesis time from 72 h to 24 h. The incorporation of organoclay in the composite film reduces the polymer chain’s mobility, and thus, increases its glass temperature. Moreover, the presence of clays increases the thermal stability of composite films. Moreover, it is shown that the mechanical properties of composite films are improved by the presence of organoclay, indicating a good dispersion of the clay inside the film and the formation of a strong interaction between the organoclay and the linseed oil polymer. It must be noted that the effect of cinnamaldehyde on the mechanical properties is less pronounced. It is observed that a part of the organoclay is dispersed inside the film and a part is located on the film surface. The latter influences the contact angle and the surface free energy value. Indeed, organoclay reduces the surface hydrophobicity, while it is observed that cinnamaldehyde has the opposite effect. For vapor water permeability (VWP), the effect of cinnamaldehyde on the film barrier properties is much higher in the presence of organoclay. It is found that the incorporation of hydrophobic compounds into the polymer films reduces the water content, which acts as a plasticizer, and therefore, decreases the WVP. The prepared composite films have a higher antioxidant activity, which increases with the amount of organoclay and cinnamaldehyde. This kind of composite is envisaged for fatty and dry food packaging applications.

## Figures and Tables

**Figure 1 molecules-27-08089-f001:**
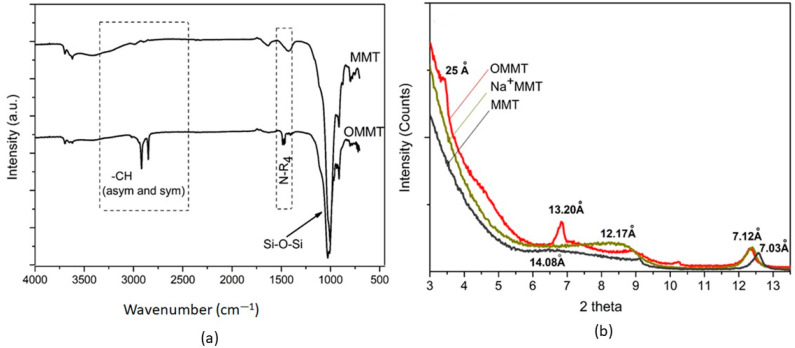
Infrared spectra of (**a**) MMT and OMMT; (**b**) X-ray diffraction patterns of MMT, Na^+^MMT, and OMMT.

**Figure 2 molecules-27-08089-f002:**
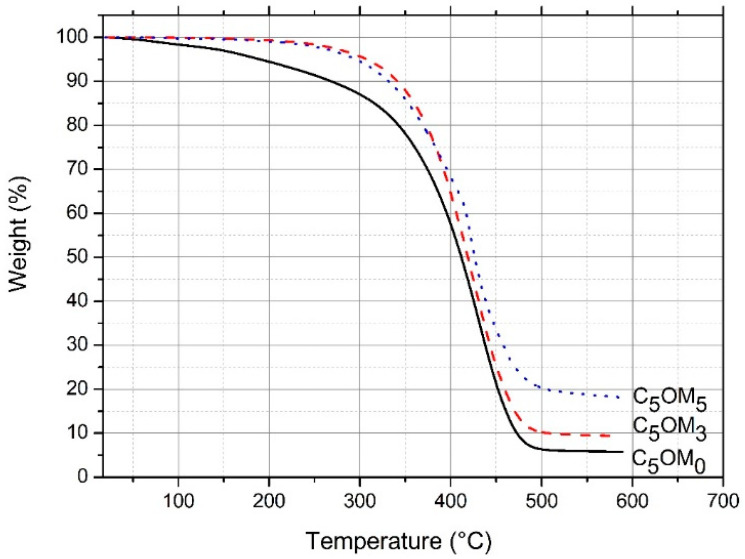
TGA thermograms of composite films (C_5_OM_0_, C_5_OM_3_, and C_5_OM_5_) under N_2_ flux (temperature rate 20 °C/min).

**Figure 3 molecules-27-08089-f003:**
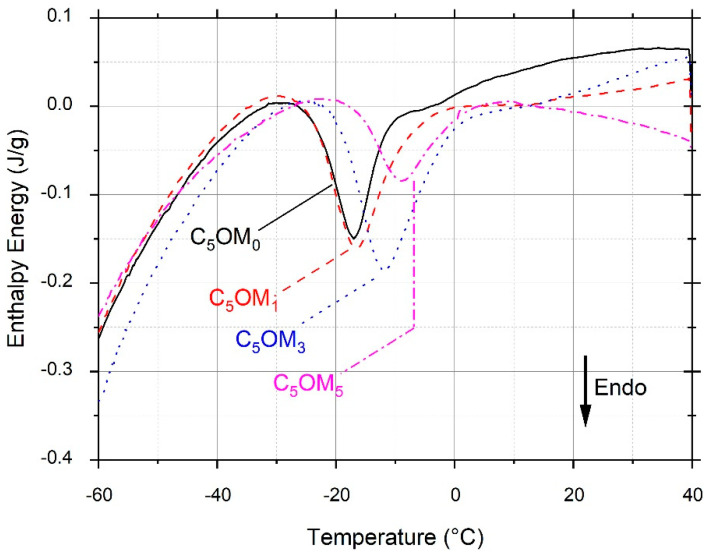
DSC Curves of C_5_OM_0_, C_5_OM_1,_ C_5_OM_3_, and C_5_OM_5_ (first heating after the ageing step of 17 h, according to the procedure presented in Section 3.3.8.

**Figure 4 molecules-27-08089-f004:**
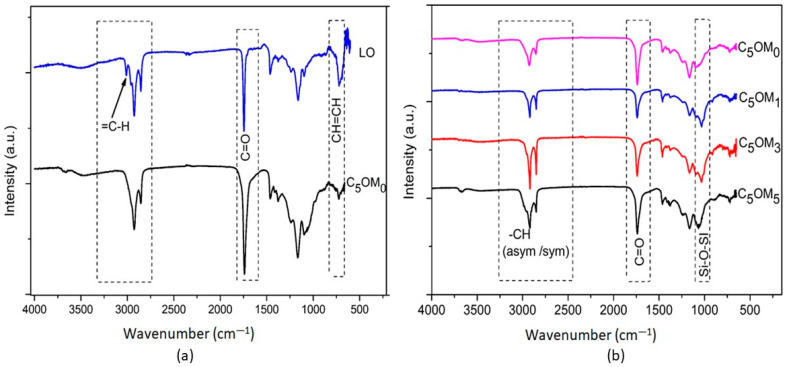
Infrared spectra of (**a**) C_5_OM_0_ and linseed oil LO; (**b**) C_5_OM_0_, C_5_OM_1_, C_5_OM_3_, and C_5_OM_5_.

**Figure 5 molecules-27-08089-f005:**
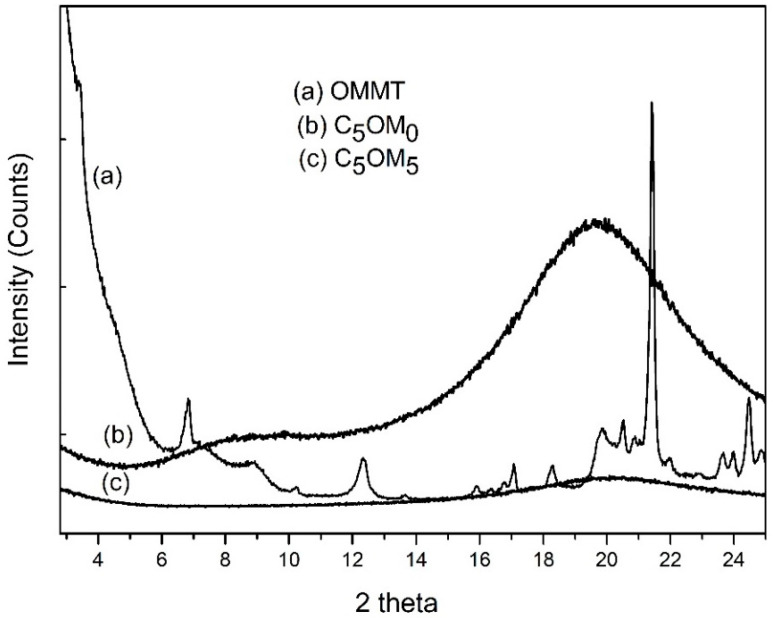
X-ray diffraction patterns of (a) C_5_OM_0_, (b) OMMT, and (c) C_5_OM_5_.

**Figure 6 molecules-27-08089-f006:**
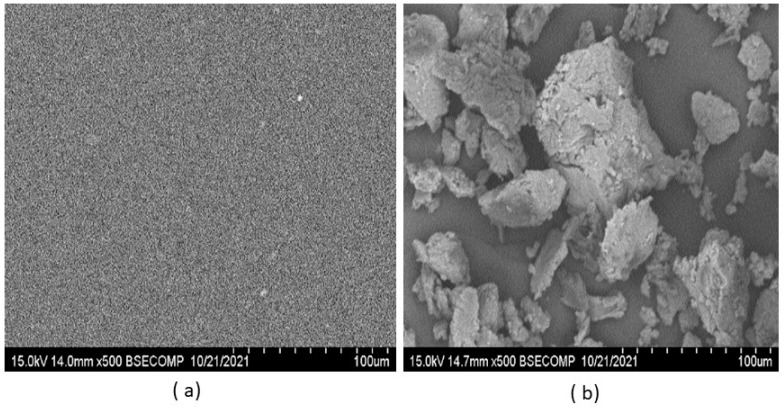
Scanning electron microscopy of (**a**) C_5_OM_0_ and (**b**) C_5_OM_5_.

**Figure 7 molecules-27-08089-f007:**
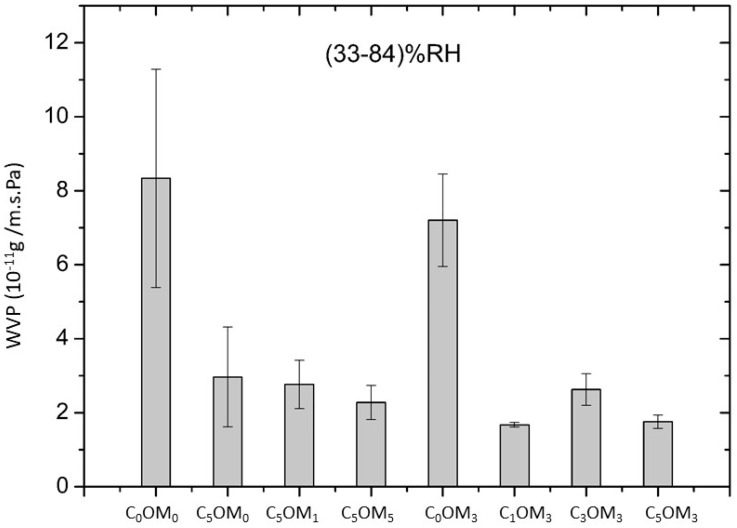
Water vapor permeability (WVP) of composite films.

**Figure 8 molecules-27-08089-f008:**
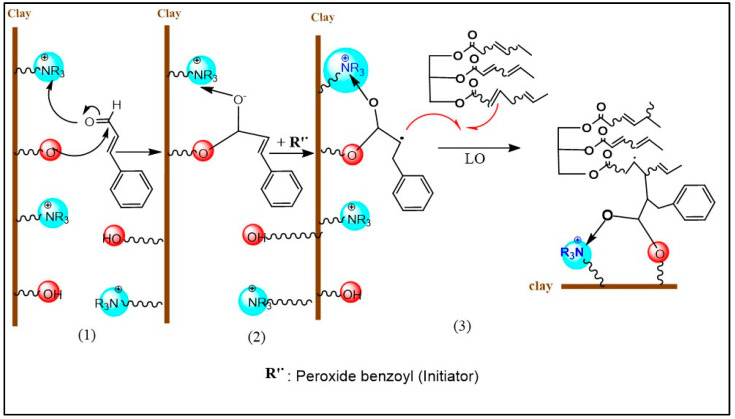
Proposed mechanism of the composite polymer formation.

**Figure 9 molecules-27-08089-f009:**
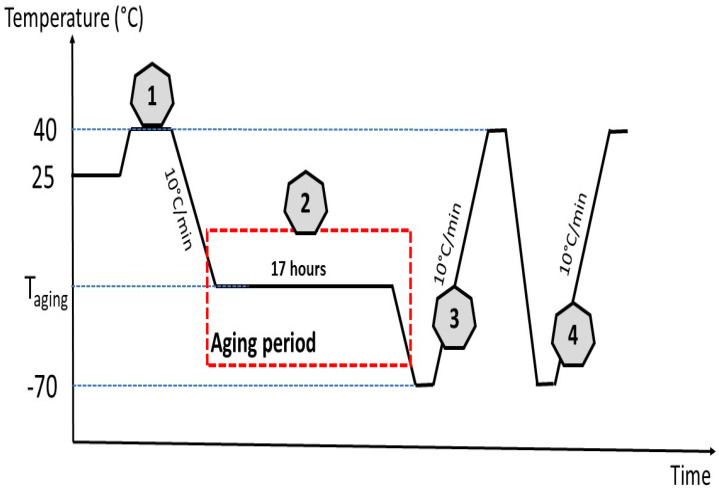
Thermograms were obtained according to this process: (1) allows erasure of the film’s thermal history; (2) maintaining the film at T_ageing_ allows “physical ageing”, and the crystallization phenomena will occur (3); the first heat with an endothermic peak; (4) allows determining the glass transition temperatures.

**Table 1 molecules-27-08089-t001:** Mechanical and surface properties of composite films.

Sample Designation	Elongation at Break (%)	Tensile Strength (MPa)	Young’s Modulus (MPa)	Contact Angle (θ. deg)	Surface Energy (mN/m)	Polar Component (mN/m)	Dispersive Component (mN/m)
C_0_OM_0_	21.2 ± 2.4 ^a^	1.3 ± 0.2 ^a^	8.6 ± 1.5 ^a^	86.9 ± 4.3 ^a^	21.2 ± 1.7 ^a^	11.7 ± 1.2 ^a^	9.6 ± 0.5 ^a^
C_5_OM_0_	26.2 ± 3.6 ^a^	0.8 ± 0.2 ^a^	4.0 ± 0.7 ^b^	88.3 ± 3.6 ^a^	22.9 ± 1.9 ^a^	8.7 ± 1.1 ^b^	14.1 ± 0.7 ^b^
C_5_OM_1_	12.8 ± 2.2 ^b^	6.6 ± 1.4 ^b^	156.0 ± 3.0 ^c^	86.7 ± 4.3 ^a^	23.9 ± 3.5 ^a,d^	9.5 ± 2.3 ^a,b^	14.4 ± 1.2 ^b^
C_5_OM_5_	1.1 ± 0.1 ^c^	9.8 ± 1.6 ^b,c^	735.1 ± 81.7 ^d^	74.0 ± 2.0 ^b,d^	31.2 ± 2.9 ^b^	21.0 ± 2.3 ^c^	10.3 ± 0.7 ^a^
C_0_OM_3_	5.3 ± 1.1 ^d^	12.4 ± 2.9 ^c^	421.9 ± 124.1 ^e^	65.5 ± 3.0 ^c^	37.45 ± 1.7 ^c^	29.4 ± 1.4 ^d^	8.0 ± 0.3 ^c^
C_3_OM_3_	8.3 ± 4.2 ^d^	10.5 ± 0.9 ^c^	288.9 ± 22.9 ^f^	70.6 ± 2.9 ^d^	35.5 ± 5.65 ^c, b^	26.8 ± 4.6 ^d,c^	8.7 ± 1.1 ^c^
C_5_OM_3_	6.4 ± 1.7 ^d^	9.8 ± 1.6 ^c^	256.2 ± 27.7 ^f^	77.5 ± 3.0 ^d^	27.8 ± 0.5 ^d^	16.2 ± 0.36 ^e^	11.7 ± 0.1 ^d^

^a–f^ Values with the same superscript letters are not significantly different at *p* level = 0.05 in each column.

**Table 2 molecules-27-08089-t002:** Antioxidant activity of composite films after 2 h and 24 h.

Sample Designation	AA (%)	AA (%)
	After 2 h	After 24 h
C0OM0	52.6 ± 2.8 ^a^	94.6 ± 1.3 ^a^
C5OM0	58.0 ± 2.6 ^b^	90.5 ± 0.8 ^b^
C5OM1	28.7 ± 0.3 ^c^	70.7 ± 0.5 ^c^
C5OM3	47.7 ± 0.1 ^d^	77.4 ± 2.0 ^d^
C5OM5	49.1 ± 1.1 ^d,e^	82.1 ± 0.5 ^e^
C0OM3	49.6 ± 0.0 ^d,e^	81.5 ± 0.0 ^e^
C1OM3	49.2 ± 1.7 ^d,e^	81.5 ± 0.4 ^e^
C3OM3	50.2 ± 0.2 ^e^	82.3 ± 0.1 ^e^
1% OMMT	1.51 ± 0.08 ^f^	2.60 ± 0.18 ^f^
3% OMMT	2.2 ± 0.49 ^f^	10.33 ± 0.56 ^g^
5% OMMT	6.67 ± 0.18 ^g^	13.46 ± 0.58 ^h^
1% Cin	1.36 ±0.45 ^f^	2.00 ± 0.71 ^f^
3% Cin	1.34 ± 1.24 ^f^	3.11 ± 0.59 ^f^
5% Cin	1.61 ± 1.70 ^f^	3.22 ± 0.51 ^f^

^a–h^ Values with the same superscript letters are not significantly different at *p* level = 0.05 in each column).

**Table 3 molecules-27-08089-t003:** Detailed composition of the polymer samples prepared from linseed oil.

Samples	% Cinnamaldehyde(*w/w*)	% Linseed Oil	% Peroxide Benzoyl (*w/w*)	% OMMT (*w/w*)	Time (h)	Image
C_0_OM_0_	0	99.8	0.2	0	72	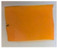
C_5_OM_0_	5	94.8	0.2	0	48	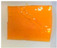
C_5_OM_1_	5	93.8	0.2	1	48	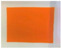
C_5_OM_3_	5	91.8	0.2	3	24	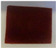
C_5_OM_5_	5	89.8	0.2	5	24	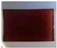
C_0_OM_3_	0	96.8	0.2	3	24	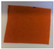
C_1_OM_3_	1	95,8	0.2	3	24	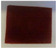
C_3_OM_3_	3	93.8	0.2	3	24	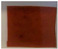

## Data Availability

The data are available on request to the corresponding author.

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
