# Peer review of "In Situ Polymerization of Linseed Oil-Based Composite Film: Enhancement of Mechanical and Water Barrier Properties by the Incorporation of Cinnamaldehyde and Organoclay"

_molecules, 2022, doi:10.3390/molecules27228089_

Round 1
Reviewer 1 Report
This paper produces linseed oil-based composite films have been prepared with cinnamaldehyde using a modified clay (organoclay), through in-situ polymerization. It is difficult to read the polymerization, since it is not explained how it occurs. Observing this difficulty and until the authors explain this process, this paper cannot be accepted. Unless they make the pertinent corrections and explain the polymerization process and preparation of the composite film.
Author Response
First, we would like to thank the three reviewers for their comments and proposal of corrections. All remarks were taken into account. Modifications were necessary and they were performed to obtain a better understanding of our results and to provide ‘clear’ explanations to the reader. We hope that changes suit now with recommendations. All comments have been considered and the text was changed according to the reviewers’ requests. Our answers are in the pdf file.

Reviewer 2 Report
Dear Author,
Study presented is quite interesting, since it entails a connection between the change of linseed oil film macroscopic properties and the addition of different components to it. Besides, several valuable data are shown. However, some changes are suggested in order to improve the comprehension and clarity of the article.
General comments:
1. It is recommended modifying surface characterization inspection. Figure 6 presents a range that can be observed using a conventional optical microscope. By showing several optical zooms, it could also be possible to check surface homogeneity. Furthermore, if the use of Nomarski microscopy or just polarized optical microscopy is feasible, valuable information regarding layer arrangement might be obtained.
2. Electron microscopy images could be very important, but at higher magnification than shown. For instance, at a nanoscopic level it may be possible to detect emulsion typical microscopic (or nanoscopic) structures, what can explain some of the macroscopic results analyzed.
3. In Materials & Methods section, SEM description is missing. It would be important to know if a metal coating was necessary.
4. If possible, it would be interesting to measure the optical density and the colorimetry of all samples, since Table 3 displays clear macroscopic differences between them.
5. If it is disclosed that temperature may alter the crystallinity (162-166, 171, and 172 lines), why not measuring the XRD of different crystalline samples and compare the results among them?
6. Appearance of the data should be organized better. For instance, CxOMx sample description appears in line 442 and 443. However, it is called since line 148, without knowing the meaning of it. Another example is the reference to Figure 8 in the Figure 3 caption.
7. Understanding that "Results" section not only includes the results, but also the "Discussion" section, it would be appraisal to enhance the connection between the subsections. It looks like different techniques were applied independently, and then put together. A linkage between each one of them would help the reader to follow the text and to know why these techniques were chosen instead of other ones. Besides, if main idea concerns the enhancement of mechanical and water barrier properties due to the cinnamaldehyde and organoclay incorporation, "Results and Discussion" should be focused on this issue throughout the whole section.
8. Another recommendation entails the use of the literature. It would be desirable to limit the number of references to the required ones, and target those that comprehend the main idea. Just an example: first reference is only a single study involving composite packaging materials, wouldn't it be better to cite a review in this case? In other parts of the text, instead, a missing reference is required.
9. A final general suggestion is related to language. Some sentences are hard to understand. Shortening them and describing the idea plainly, directly and clearly might help. Besides, a whole typographic/spelling revision is recommended.
In the following, some specific comments of parts of the text are outlined:
[22] It is not clear in the abstract the reason of adding cinnamaldehyde to the film. First appearance of cinnamaldehyde in the article is in the sentence "...whatever the amount of cinnamaldehyde.", without previously explaining why this component was chosen.
[41-42] "However, the low mechanical properties, water, or gas barrier properties are the main obstacles to the development of bio-sourced materials". It depends on the required applicability. For some utilities it might be preferably low mechanical properties.
[64-75] Consider improving the grammar of the text.
[111] Section "Results and Discussion".
[125] Acronyms (MMT and OMMT) are described above this line. There is no need to repeat the acronym definition.
[211] "as confirmed by the elementary analysis of these particles". If a SEM-EDX was performed, data should appear. If "elementary" does not imply elements, but something else, the sentence should be modified.
[214] "can be considered the tip of the “iceberg”". If there are data that prove it, it would be convenient to show.
[229-230, 233-236] "This decrease could be explained by the hydrophobic character of cinnamaldehyde which reduces water transport", "In general, the incorporation of hydrophobic compounds into the hydrophilic or hydrophobic polymer films reduces the moisture content which is responsible for the plasticity of the film since water act as a plasticizer and therefore increases the WVP". This requires an explanation or a reference. Linseed oil is hydrophobic too, why adding cinnamaldehyde VWP behavior changes? And, why WVP of C3OM3 is higher than C1OM3?
[237] Why cinnamaldehyde addition (4 ones on the right) does not change gradually the WVP?
[347] Is it a GC-MS? If so, data could be of interest of many readers. It is suggested including them in the article, perhaps in the Supporting Information section.
[358-360] "As reported by several studies, linseed oil showed higher antioxidant activity in the form of hydroxyl and superoxide radical scavenging". Reference(s) should be added.
[364-370] Including pure cinnamaldehyde and pure organoclay data in Table 2 might help to compare them.
[425-426] "This clay fraction has a fine-particle structure with particle size in the order of 2 μm." How this particular value is known?
Best regards.
Author Response

(The authors gave the same response as above.)

Reviewer 3 Report
The manuscript is well organized and the topic is interesing for wide audience. However, some results deserve better and ore precise explanations. I recommend the paper for publication after the following corrections:
a) line 142-the authors said "the composite materials contain progressive amount of clay"...it is not progressive amount, please re-phrase this
b) line 145-"an increase in degradation temperature..."-what is degradation temperature-onset temperature or the temperature of maximum degradtion rate, please write this clear...also, the temperature did not increase, but was shifted to higher values
c) line 146-" the amount of inorganic compound increased in the same way as the amount of organoclay introduced..." this sentence is not written clear...how this is verfified that you have increased amount of clay in your samples by tga, you need to clarify that better. i suppose you came to this conclusion according to your char residue. also, pay attention on tenses you used for explanation of your results. You are mixing the present tense and past tense all the time. You write all results or in present tense, or in past tense, but not mixing it.
d) line 212-wrong conclusion for the sem morphology of your sample. figure 6b clearly shows that there is no imbided clay into film, how you explained, but the agglomeration and not good dispersion into polymer matrix at all.
e) english grammar requires editing and improvement.
Author Response

(The authors gave the same response as above.)

Round 2
Reviewer 1 Report
The authors have adequately responded to requests to review this paper, therefore it should be published in Molecules.
Author Response
Thank you for your comments.
Regards
Reviewer 2 Report
Dear Author,
Thanks for the modifications performed. However, there are still some changes/refinements that should be considered.
General comments:
1. It is recommended modifying surface characterization inspection. Figure 6 presents a range that can be observed using a conventional optical microscope. By showing several optical zooms, it could also be possible to check surface homogeneity. Furthermore, if the use of Nomarski microscopy or just polarized optical microscopy is feasible, valuable information regarding layer arrangement might be obtained.
Answer: Thank you for this remark. We don’t use Nomarski microscopy as this later is specially used to analyse planar semiconductor processing. In our case, the objective is to understand how the organoclay is dispersed or imbibed in the composite film. An observation using scanning electron microscopy (Hitachi SU-1510 microscope with an accelerating voltage of 20 keV) seems to be sufficient to observe these changes.
Comment refers to optical microscopy images (even conventional ones). It is still considered important to show these optical microscopy images in the article, since they can permit illustrate homogeneity much better than coated electron microscopy (EM) images. EM images focus on small regions and record only parts presenting high electron reflection, which is dependent on the coating. Hence, white light optical microscopy in reflection at different scales might provide even more information than the EM images presented. Furthermore, optical images at different scales may help to support better what is stated in the next comment’s answer: “the clay clusters don’t cover all the film”.
A conventional optical microscope is a common scientific tool in almost every laboratory, and white light reflection images are very straightforward. Insertion shall reinforce the article.
Regarding Nomarski (or DIC) optics, it is widely used in biology and permits characterizing organic structures. It is not limited to “planar semiconductor processing”. In general, polarized microscopy is particularly useful for detecting crystalline structures. In the manuscript it is suggested that there are crystalline structures constituting the films, that is the reason why this technique was suggested.
2. Electron microscopy images could be very important, but at higher magnification than shown. For instance, at a nanoscopic level it may be possible to detect emulsion typical microscopic (or nanoscopic) structures, what can explain some of the macroscopic results analyzed.
Answer: Thank you for this remark. We have performed observations at higher magnification (x3000) (see figure 1). Without clay (C5OM0), the film has a homogeneous surface as can be seen in (Figure 6a with low magnification x500). For the composite films (C5OM5), increasing the magnitude of the microscope allow us to see better the different layers of the clay structure. However, we decided to keep figure 6(b) with low magnification (x 500) in order to show that the clay clusters don’t cover all the film.
Unfortunately, EM images recorded with this microscope at higher magnification are blurred, perhaps due to spherical aberration. Obviously, it is better to avoid the unfocused images. Scale is still quite big even for optical images (typical maximum resolution below 1 µm). Maybe a focused optical inspection permits detecting typical microscopic emulsion structures.
Specific comments:
[42-43] "However, the low mechanical properties, water, or gas barrier properties are the main obstacles to the development of bio-sourced materials". It depends on the required applicability. For some utilities it might be preferably low mechanical properties.
Answer: We totally agree with this remark. The mechanical properties depend on the application. For films which need to be degraded easily, the low mechanical property is recommended. However, the real challenge in food packaging is to increase this property.
“Food packaging” includes quite flexible films too, which is also challenging. If objective is restricted to the case of rigid containers, it should be clarified in the text.
[215] "can be considered the tip of the “iceberg”". If there are data that prove it, it would be convenient to show.
Answer: The “iceberg” term explains the fact that the clay is both on the surface and inside the composite film. The data which convince us about this organization are: the amount of clay in the composite film is higher (Figure 2 TGA), the surface is less hydrophobic due to the presence of the clay on the surface, and the clay is inside because the mechanical properties were changed.
A higher amount of clay, a reduction of hydrophobicity, and a change in the mechanical properties do not guarantee an “iceberg” conformation. There could be loose debris on the surface and clay pieces deep inside the material (not saying this is the case), and same effects may occur. In case the sentence is speculative, it should be declared like that. Otherwise, it might be confused to the reader, since reasons provided do not really demonstrate the fact. If required, it may be proved by taking transversal EM images (i.e., captures on a slice of the film).
[229-230, 233-236] "This decrease could be explained by the hydrophobic character of cinnamaldehyde which reduces water transport", "In general, the incorporation of hydrophobic compounds into the hydrophilic or hydrophobic polymer films reduces the moisture content which is responsible for the plasticity of the film since water act as a plasticizer and therefore increases the WVP". This requires an explanation or a reference. Linseed oil is hydrophobic too, why adding cinnamaldehyde VWP changes the behavior? And, why WVP of C3OM3 is higher than C1OM3?
Answer: Thank you for this interesting remark. In figure 7, the incorporation of Cin decreases the WVP explained by the hydrophobic character of cin. In addition, Cin participates in the polymerization reaction (Figure 8), the polymer network becomes denser, and this is reduced the water transfer. Changes made in the revised manuscript (see lines 236-239)
Only hydrophobicity cannot explain the fact, because linseed oil is hydrophobic too. Densification of the network does, but this is not proved. It can be hypothesized, but not declared as true without evidence.
Consider reviewing the added text. It does not seem to be correctly written.
[239] Why cinnamaldehyde addition (4 ones on the right) does not change gradually the WVP?
Answer: Thank you for this remark which is difficult to interpret. Without OMMT, the effect of Cin is to reduce the WVP. However, when the clay is added the system becomes complex and it is difficult to understand the effect of both. Indeed, OMMT and Cin participate in the polymerization reaction and maybe there is an optimum amount of both to obtain denser films.
For this reason, I suggested to measure the optical density. In the corresponding answer, it was indicated that “the thickness of the films can vary”. WVP depends on the thickness, was this parameter controlled during the measurements?
A possible explanation should be added to the manuscript, since this result is puzzling to the reader. Rationalization through the increase of the density is understandable, but it has not been demonstrated. It might be postulated, but not affirmed.
[349] Is it a GC-MS? If so, data could be of interest of many readers. It is suggested including them in the article, perhaps in the Supporting Information section.
Answer: We agree with the comment. The GC-MS analysis was done but we don’t see any additional information for the reason that this analysis was not added to the manuscript.
In the text it is maintained the following: “This high value of AA is related to its higher fatty acid content. GC-MC analysis of linseed oil (data not shown here) displayed that this oil has three principal fatty acids: 22% of C18:1, 15% of C18:2, and 50% of 350 C18:3.” Firstly, it is written “GC-MC” instead of “GC-MS”. Secondly, there is no need of “additional information”: it is stated that these data permits determining fatty acid concentration, which is enough to justify their incorporation into the article, at least in the Supporting Information.
[427-428] "This clay fraction has a fine-particle structure with particle size in the order of 2 μm." How this particular value is known?
Answer: Clay minerals are a diverse group of hydrous layer aluminosilicates that constitute the greater part of the phyllosilicate family of minerals. They are commonly defined by geologists as hydrous layer aluminosilicates with a particle size < 2 microns. Separation is generally done by successive sedimentations from a suspension of dispersed clay.
It should be supported by measurements or a reference. However, even if it were proved elsewhere that size is below 2 microns, it does not imply that size is of the order of 2 µm. They can have a “size < 2 microns” and range much less than 2 µm.
Best regards.
Author Response
Dear Reviwer,
Thank you for your interesting comments which for sure will improve the quality of our paper. We have also corrected the English and the style by a native english professional. Spelling and errors have incorporated directly in the revised manuscript without any notification. However, the main modifications due to the reviewer questions are highlighted in yellow color in the revised manuscript.
The answers to the round 2 questions (bleue color) are presented in the following document with our answers (green color).
Regards
